# The nonequilibrium cost of accurate information processing

Giulio Chiribella [1,2,3] ✉, Fei Meng [1,4], Renato Renner [5] & Man-Hong Yung[4,6]

Accurate information processing is crucial both in technology and in nature. To achieve it, any information processing system needs an initial supply of resources away from thermal equilibrium. Here we establish a fundamental limit on the accuracy achievable with a given amount of nonequilibrium resources. The limit applies to arbitrary information processing tasks and arbitrary information processing systems subject to the laws of quantum mechanics. It is easily computable and is expressed in terms of an entropic quantity, which we name the reverse entropy, associated to a time reversal of the information processing task under consideration. The limit is achievable for all deterministic classical computations and for all their quantum extensions. As an application, we establish the optimal tradeoff between nonequilibrium and accuracy for the fundamental tasks of storing, transmitting, cloning, and erasing information. Our results set a target for the design of new devices approaching the ultimate efficiency limit, and provide a framework for demonstrating thermodynamical advantages of quantum devices over their classical counterparts.

Many processes in nature depend on accurate processing of information. For example, the development of complex organisms relies on the accurate replication of the information contained in their DNA, which takes place with an error rate estimated to be less than one basis per billion[1].

At the fundamental level, information is stored into patterns that stand out from the thermal fluctuations of the surrounding environment[2,3]. In order to achieve deviations from thermal equilibrium, any information-processing machine needs an initial supply of systems in a non-thermal state[4,5]. For example, an ideal copy machine for classical data requires at least a clean bit for every bit it copies[6–8]. For a general information-processing task, a fundamental question is: what is the minimum amount of nonequilibrium needed to achieve a target level of accuracy? This question is especially prominent at the quantum scale, where many tasks cannot be achieved perfectly even in principle, as illustrated by the no-cloning theorem[9,10].

In recent years, there has been a growing interest in the interplay between quantum information and thermodynamics[11–13], motivated both by fundamental questions[14–18] and by the experimental realisation of new quantum devices[19–21]. Research in this area led to the development of resource-theoretic frameworks that can be used to study thermodynamics beyond the macroscopic limit[22–30]. These frameworks have been applied to characterise thermodynamically allowed state transitions, to evaluate the work cost of logical operations[31,32] and to study information erasure and work extraction in the quantum regime[33–35]. From a different perspective, relations between accuracy and entropy production have been investigated in the field of stochastic thermodynamics[36–40], referring to specific physical models such as classical Markovian systems in nonequilibrium steady states.

Here, we establish a fundamental tradeoff between accuracy and nonequilibrium, valid at the quantum scale and applicable to arbitrary information-processing tasks. The main result is a limit on the

[1]QICI Quantum Information and Computation Initiative, Department of Computer Science, The University of Hong Kong, Pokfulam Road, Hong Kong SAR, China. [2]Department of Computer Science, University of Oxford, Parks Road, Oxford OX1 3QG, UK. [3]Perimeter Institute for Theoretical Physics, 31 Caroline Street North, Waterloo, ON N2L 2Y5, Canada. [4]Department of Physics, Southern University of Science and Technology, 518055 Shenzhen, China. [5]Institute for Theoretical Physics, ETH Zürich, Zürich, Switzerland. [6]Shenzhen Key Laboratory of Quantum Science and Engineering, 518055 Shenzhen, China. ✉e-mail: giulio@cs.hku.hk

accuracy, expressed in terms of an entropic quantity, which we call the reverse entropy, associated with a time reversal of the information-processing task under consideration. The limit is attainable in a broad class of tasks, including all deterministic classical computations and all quantum extensions thereof. For the task of erasing quantum information, our limit provides, as a byproduct, the ultimate accuracy achievable with a given amount of work. For the tasks of storage, transmission, and cloning of quantum information, our results reveal a thermodynamic advantage of quantum setups over all classical setups that measure the input and generate their output based only on the measurement outcomes. In the cases of storage and transmission, we show that quantum machines can break the ultimate classical limit on the amount of work required to achieve a desired level of accuracy. This result enables the demonstration of work-efficient quantum memories and quantum communication systems outperforming all possible classical setups.

## Results

### The nonequilibrium cost of accuracy

At the most basic level, the goal of information processing is to set up a desired relation between an input and an output. For example, a deterministic classical computation amounts to transforming a bit string $x$ into another bit string $f(x)$, where $f$ is a given function. In the quantum domain, information-processing tasks are often associated with ideal state transformations $\rho_x \mapsto \rho_x'$, in which an input state described by a density operator $\rho_x$ has to be converted into a target output state described by another density operator $\rho_x'$, where $x$ is a parameter in some given set X.

Since every realistic machine is subject to imperfections, the physical realisations of an ideal information-processing task can have varying levels of accuracy. Operationally, the accuracy can be quantified by performing a test on the output of the machine and by assigning a score to the outcomes of the measurement. The resulting measure of accuracy is given by the expectation value of a suitable observable $O_x$, used to assess the closeness of the output to the target state $\rho_x'$. In the worst case over all possible inputs, the accuracy achieved in a given task $\mathcal{T}$ has the expression $\mathcal{F}_{\mathcal{T}}(\mathcal{M}) = \min_x \mathrm{Tr}[O_x \mathcal{M}(\rho_x)]$, where $\mathcal{M}$ is the quantum channel (completely positive trace-preserving map) describing the action of the machine. Here, the dependence of the input states $\rho_x$ and output observables $O_x$ on the parameter $x$ is fully general, and includes in particular cases where multiple observables are tested for the same input state. The range of values for the function $\mathcal{F}_{\mathcal{T}}$ depends on the choice of observables $O_x$: for example, if all the observables $O_x$ are projectors, the range of $\mathcal{F}_{\mathcal{T}}$ will be included in the interval $[0, 1]$.

Accurate information processing generally requires an initial supply of systems away from equilibrium. The amount of nonequilibrium required to implement a given task can be rigorously quantified in a resource-theoretic framework where Gibbs states are regarded as freely available, and the only operations that can be performed free of cost are those that transform Gibbs states into Gibbs states[28,32]. These operations, known as Gibbs preserving, are the largest class of processes that maintain the condition of thermal equilibrium. The initial nonequilibrium resources can be represented in a canonical form by introducing an information battery[31,32], consisting of an array of qubits with degenerate energy levels. The battery starts off with some qubits in a pure state (hereafter called the clean qubits), while all the remaining qubits are in the maximally mixed state. To implement the desired information-processing task, the machine will operate jointly on the input system and on the information battery, as illustrated in Fig. 1.

The number of clean qubits required by a machine is an important measure of efficiency, hereafter called the nonequilibrium cost. For a given quantum channel $\mathcal{M}$, the minimum nonequilibrium cost of any machine implementing channel $\mathcal{M}$ (or some approximation thereof) has been evaluated in refs. 31, 32. Many information-processing tasks,

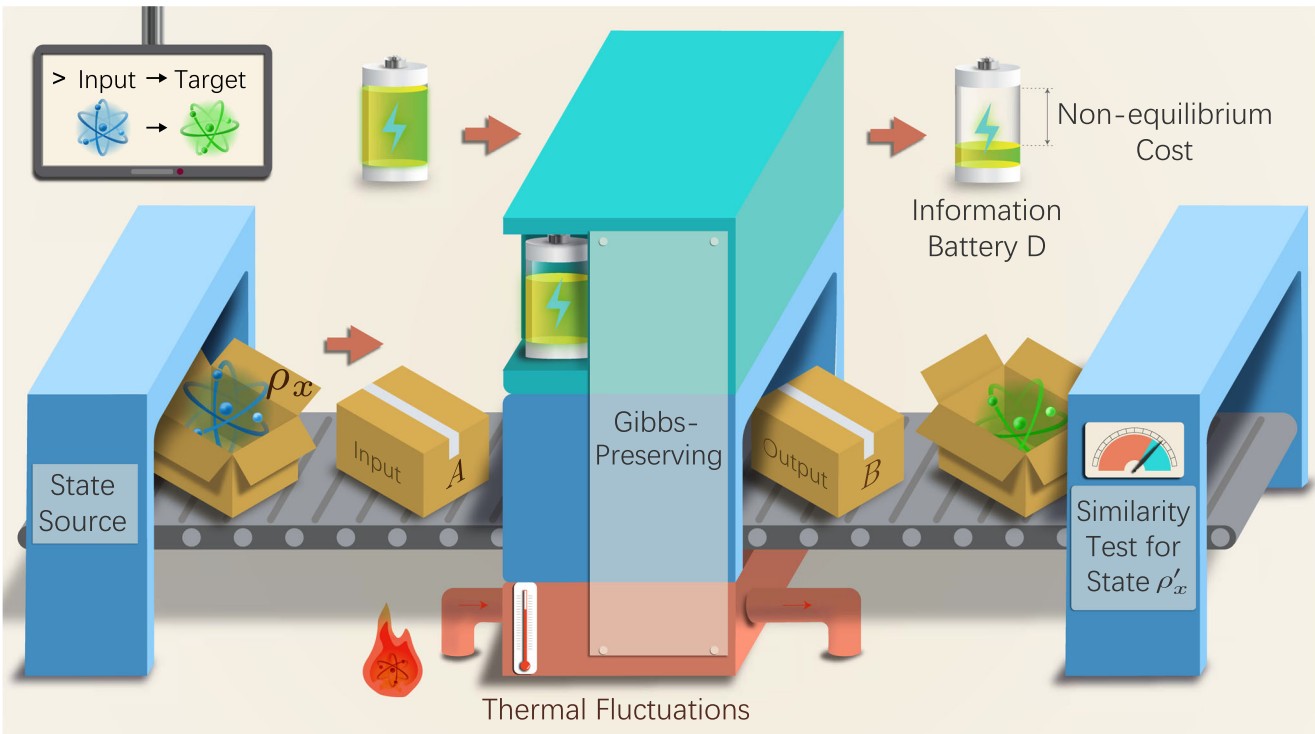

**Fig. 1 | The nonequilibrium cost of accuracy.** A source generates a set of input states for an information-processing machine. The machine uses an information battery (a supply of qubits initialised in a fixed pure state) and thermal fluctuations (a reservoir in the Gibbs state) to transform the input state $\rho_x$ into an approximation of the ideal target states $\rho_x'$. Finally, the similarity between the output and the target states is assessed by a measurement. The number of pure qubits consumed by the machine is the nonequilibrium cost that needs to be paid in order to achieve the desired level of accuracy.

however, are not uniquely associated with a specific quantum channel: for example, most state transitions $\rho \mapsto \rho'$ can be implemented by infinitely many different quantum channels, which generally have different costs. When a task can be implemented perfectly by more than one quantum channel, the existing results do not identify, in general, the minimum nonequilibrium cost that has to be paid for a desired level of accuracy. Furthermore, there also exist information-processing tasks, such as quantum cloning[9,10], that cannot be perfectly achieved by any quantum channel. In these scenarios, it is important to establish a direct relation between the accuracy achieved in the given task and the minimum cost that has to be paid for that level of accuracy. Such a relation would provide a direct bridge between thermodynamics and abstract information processing, establishing a fundamental efficiency limit valid for all machines allowed by quantum mechanics.

In this paper, we build concepts and methods for determining the nonequilibrium cost of accuracy in a way that depends only on the information-processing task under consideration, and not on a specific quantum channel. Let us denote by $c(\mathcal{M}, \Pi_A)$ the nonequilibrium cost required for implementing a given channel $\mathcal{M}$ on input states in the subspace specified by a projector $\Pi_A$. We then define the nonequilibrium cost for achieving accuracy $F$ in a task $\mathcal{T}$ as $c_{\mathcal{T}}(F) := \min\{c(\mathcal{M}, \Pi_A) \mid \mathcal{F}_{\mathcal{T}}(\mathcal{M}) \geq F\}$. Note that the the specification of the input subspace is included in the task $\mathcal{T}$. In the following we focus on tasks where the input subspace is invariant under time evolution, namely $[\Pi_A, H_A] = 0$, where $H_A$ is the Hamiltonian of the input system. Our main goal will be to evaluate $c_{\mathcal{T}}(F)$, the nonequilibrium cost of accuracy.

In Methods, we provide an exact expression for $c_{\mathcal{T}}(F)$. The expression involves a semidefinite programme, which can be solved numerically for low-dimensional systems, thus providing the exact tradeoff between nonequilibrium and accuracy. Still, brute-force optimisation is intractable for high-dimensional systems. For this reason, it is crucial to have a computable bound that can be applied in a broader range of situations. The central result of the paper is a universal bound, valid for all quantum systems and to all information-processing tasks: the bound reads

$$c_{\mathcal{T}}(F) \geq \kappa_{\mathcal{T}} + \log F, \qquad (1)$$

where $\kappa_{\mathcal{T}} := -\log F_{\max}^{\mathcal{T}_{\text{rev}}}$ is an entropic quantity, hereafter called the reverse entropy, and $F_{\max}^{\mathcal{T}_{\text{rev}}}$ is the maximum accuracy allowed by quantum mechanics to a time-reversed information-processing task $\mathcal{T}_{\text{rev}}$, precisely defined in the following section (see Supplementary Note 1 for the derivation of Eq. (1)). Note that the reverse entropy is a monotonically decreasing function of $F_{\max}^{\mathcal{T}_{\text{rev}}}$, and becomes zero when the time-reversed task can be implemented with unit accuracy.

Eq. (1) can be equivalently formulated as a limit on the accuracy attainable with a given budget of nonequilibrium resources: for a given number of clean qubits $c$, the maximum achievable accuracy in the task $\mathcal{T}$, denoted by $F_{\mathcal{T}}(c) := \max\{\mathcal{F}_{\mathcal{T}}(\mathcal{M}) \mid c(\mathcal{M}, \Pi_A) \leq c\}$, satisfies the bound

$$F_{\mathcal{T}}(c) \leq 2^{c - \kappa_{\mathcal{T}}}. \qquad (2)$$

This bound represents an in-principle limit on the performance of every information-processing machine. The bounds (1) and (2) are achievable in a number of tasks, and have a number of implications that will be discussed in the following sections.

## Time-reversed tasks and reverse entropy

Here, we discuss the notion of time reversal of an information-processing task. Let us start from the simplest scenario, involving transformations of a fully degenerate system into itself. For a state transformation task $\rho_x \mapsto \rho'_x$, we consider without loss of generality an accuracy measure for which the observables $O_x$ are positive operators, proportional to quantum states. We then define a time-reversed task $\mathcal{T}_{\text{rev}}$, where the role of the input states $\rho_x$ and of the output observables $O_x$ are exchanged. The accuracy of a generic channel $\mathcal{M}$ in the execution of the time-reversed task is specified by the reverse accuracy $\mathcal{F}_{\mathcal{T}_{\text{rev}}}(\mathcal{M}) := \min_x \text{Tr}[\rho_x \mathcal{M}(O_x)]$. Maximising over all possible channels, we obtain $F_{\max}^{\mathcal{T}_{\text{rev}}}$ and define $\kappa_{\mathcal{T}} = -\log F_{\max}^{\mathcal{T}_{\text{rev}}}$.

For systems with nontrivial energy spectrum, we define the time-reversed task in terms of a time reversal of quantum operations introduced by Crooks[41] and recently generalised in ref. [42] (this time-reversal operation is also related to Petz's recovery map)[43–45]. In the Gibbs preserving context, this time-reversal exchanges states with observables, mapping Gibbs states into trivial observables (described by the identity matrix) and vice-versa. More generally, the time-reversal maps the states $\rho_x$ into the observables $\widetilde{O}_x := \Gamma_A^{-\frac{1}{2}} \rho_x \Gamma_A^{-\frac{1}{2}}$ and the observables $O_x$ into the (unnormalised) states $\widetilde{\rho}_x := \Gamma_B^{\frac{1}{2}} O_x \Gamma_B^{\frac{1}{2}}$, where $\Gamma_A$ and $\Gamma_B$ are the Gibbs states of the input and output systems, respectively. The reverse accuracy of a channel $\mathcal{M}$ is then defined as $\mathcal{F}_{\mathcal{T}_{\text{rev}}}(\mathcal{M}) := \min_x \text{Tr}[\widetilde{O}_x \mathcal{M}(\widetilde{\rho}_x)]$. In the Methods section, we show that the reverse entropy can be equivalently written as

$$\kappa_{\mathcal{T}} = \max_{\mathbf{p}} H_{\min}(A|B)_{\omega_{\mathcal{T}, \mathbf{p}}}. \qquad (3)$$

where $\mathbf{p} = (p_x)_{x \in \mathsf{X}}$ is a probability distribution, $\omega_{\mathcal{T}, \mathbf{p}} = \sum_x p_x \Gamma_A^{-\frac{1}{2}} \rho_x^T \Gamma_A^{-1/2} \otimes \Gamma_B^{\frac{1}{2}} O_x \Gamma_B^{\frac{1}{2}}$ is an operator acting on the tensor product of the input and output systems, $\rho_x^T$ is the transpose of the density matrix $\rho_x$ with respect to the energy eigenbasis, and $H_{\min}(A|B)_{\omega_{\mathcal{T}, \mathbf{p}}} := -\log \min\{\text{Tr}[\Lambda_B] \mid (I_A \otimes \Lambda_B) \geq \omega_{\mathcal{T}, \mathbf{p}}\}$ is the conditional min-entropy[43–45].

Crucially, the reverse entropy depends only on the task under consideration, and not on a specific quantum channel used to implement the task. In fact, the reverse entropy is well-defined even for tasks that cannot be perfectly achieved by any quantum channel, as in the case of ideal quantum cloning, and even for tasks that are not formulated in terms of state transitions (see Methods).

To gain a better understanding of the reverse entropy, it is useful to evaluate it in some special cases. Consider the case of a classical deterministic computation, corresponding to the evaluation of a function $y = f(x)$. In this case the reverse entropy is

$$\kappa_f = D_{\max}(p_f \| g_B), \qquad (4)$$

where $D_{\max}(p \| q) = \max_y p(y)/q(y)$ is the max Rényi divergence between two probability distributions $p(y)$ and $q(y)$[46], $g_B(y)$ is Gibbs distribution for the output system, and $p_f(y)$ is the probability distribution of the random variable $y = f(x)$, when $x$ is sampled from the Gibbs distribution (see Supplementary Note 2 for the derivation). Eq. (4) shows that the reverse entropy of a classical computation is a measure of how much the computation transforms thermal fluctuations into states that deviate from thermal equilibrium.

In the quantum case, however, physical limits to the execution of the time-reversed task can arise even without any deviation from thermal equilibrium. Consider for example the transposition task $\rho_x \mapsto \rho_x^T$[47–52], where $x$ parametrises all the possible pure states of a quantum system. This transformation does not generate any deviation from equilibrium as it maps Gibbs states into Gibbs states. On the other hand, in the fully degenerate case the time-reversed task is still transposition, and perfect transposition is forbidden by the laws of quantum mechanics[47–52]. The maximum fidelity of an approximate transposition is $F_{\text{trans}} = 2/(d+1)$ for $d$-dimensional quantum systems, and therefore $\kappa_{\text{trans}} = \log[(d+1)/2]$.

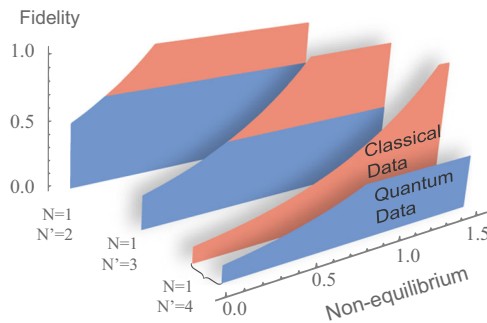

**Fig. 2 | Maximum cloning fidelity for a given amount of nonequilibrium resources.** The optimal accuracy-nonequilibrium tradeoff is depicted for $N \to N'$ cloning machines with $N = 1$ and $N' = 2,3,4$. The fidelities for copying classical (red region) and quantum data (blue region) are limited by the same boundary curve, except that the fidelity for the task of copying quantum data cannot reach to 1 due to the no-cloning theorem.

## Condition for achieving the limit

The appeal of the bounds (1) and (2) is that they are general and easy to use. But are they attainable? To discuss their attainability, it is important to first identify the parameter range in which these bounds are meaningful. First of all, the bound (1) is only meaningful when the desired accuracy does not exceed the maximum accuracy $F_{max}$ allowed by the laws of physics for the task $\mathcal{T}$. Similarly, the bound (2) is only meaningful if the initial amount of nonequilibrium resources does not go below the smallest nonequilibrium cost of an arbitrary process acting on the given input subspace, hereafter denoted by $c_{min}$. By maximising the accuracy over all quantum channels with minimum cost $c_{min}$, we then obtain a minimum value $F_{min}$ below which reducing the accuracy does not result in any reduction of the nonequilibrium cost.

We now provide a criterion that guarantees the attainability of the bounds (1) and (2) in the full interval $[F_{min}, F_{max}]$. Since the two bounds are equivalent to one another, we will focus on bound (1). The condition for attainability in the full interval $[F_{min}, F_{max}]$ is attainability at the maximum value $F_{max}$. As we will see in the rest of the paper, this condition is satisfied by a number of information-processing tasks, notably including all classical computations and all quantum extensions thereof.

**Theorem 1.** For every information-processing task $\mathcal{T}$ with $[\Pi_A, H_A] = 0$, if the bound (1) is attainable for a value of the accuracy $F_0$, then it is attainable for every value of the accuracy in the interval $[F_{min}, F_0]$, with $F_{min} = 2^{c_{min} - \kappa_{\mathcal{T}}}$. In particular, if the bound is attainable for the maximum accuracy $F_{max}$, then it is attainable for every value of the accuracy in the interval $[F_{min}, F_{max}]$.

In Supplementary Note 3, we prove the theorem by explicitly constructing a family of channels that achieve the bound (1).

By evaluating the nonequilibrium cost of specific quantum channels, one can prove the attainability of the bound (1) for a variety of different tasks. For example, the bound (1) is attainable for every deterministic classical computation. Moreover, it is achievable for every quantum extension of a classical computation: on Supplementary Note 4 we show that for every value of the accuracy, the nonequilibrium cost is the same for the original classical computation and for its quantum extension, and therefore the achievability condition holds in both cases.

The nonequilibrium cost $c_{\mathcal{T}}(F)$ provides a fundamental lower bound to the amount of work that has to be invested in order to achieve accuracy $F$. Indeed, the minimum work cost of a specific channel $\mathcal{M}$, denoted by $W(\mathcal{M}, \Pi_A)$ can be quantified by the minimum number of clean qubits needed to implement the process in a scheme like the one in Fig. 1, with the only difference that Gibbs preserving

operations are replaced by thermal operations, that is, operations resulting from a joint energy-preserving evolution of the system together with auxiliary systems in the Gibbs state[24,25]. Since thermal operations are a proper subset of the Gibbs preserving operations[28], the restriction to thermal operations generally results into a larger number of clean qubits, and the work cost is lower bounded as $W(\mathcal{M}, \Pi_A) \geq kT(\ln 2) c(\mathcal{M}, \Pi_A)$, where $k$ is the Boltzmann constant and $T$ is the temperature. By minimising both sides over all channels that achieve accuracy $F$, we then get the bound $W_{\mathcal{T}}(F) \geq kT(\ln 2) c_{\mathcal{T}}(F)$, where $W_{\mathcal{T}}(F) := \min\{W(\mathcal{M}, \Pi_A) | F_{\mathcal{T}}(\mathcal{M}) \geq F\}$ is the minimum work cost that has to be paid in order to reach accuracy $F$.

The achievability of this bound is generally nontrivial, except for operations on fully degenerate classical systems, wherein the sets of thermal operations and Gibbs preserving maps coincide due to Birkhoff's theorem[53]. Another example is the task of erasing quantum states, corresponding to the state transformation $\rho_x \mapsto |0\rangle\langle 0|$, where $\rho_x$ is an arbitrary state and $|0\rangle$ is the ground state. In Supplementary Note 4, we show that the bound $W_{erase}(F) \geq kT \ln 2 \, c_{erase}(F)$ holds with the equality sign, and the minimum work cost of approximate erasure is given by

$$W_{erase}(F) = \Delta A + kT \ln F, \tag{5}$$

where $\Delta A$ is the difference between the free energy of the ground state and the free energy of the Gibbs state, and the equality holds for every value of $F$ in the interval $[F_{min}, F_{max}]$, with $F_{min} = e^{-\Delta A/(kT)}$ and $F_{max} = 1$.

## Nonequilibrium cost of classical cloning

Copying is the quintessential example of an information-processing task taking place in nature, its accurate implementation being crucial for processes such as DNA replication. In the following, we will refer to the copying of classical information as classical cloning. In abstract terms, the classical cloning task is to transform $N$ identical copies of a pure state picked from an orthonormal basis into $N' \geq N$ copies of the same state. Classically, this corresponds to the transformation $|x\rangle\langle x|^{\otimes N} \mapsto |x\rangle\langle x|^{\otimes N'}$, where $x$ labels the vectors of an orthonormal basis. The reverse entropy can be computed from Eq. (4), which gives

$$\kappa_{clon}^{C} = \frac{\Delta N \, \Delta A_{max}}{kT \ln 2}, \tag{6}$$

where $\Delta N := N' - N$ the number of extra copies, and $\Delta A_{max}$ is the maximum difference between the free energy of a single-copy pure state and the free energy of the single-copy Gibbs state. Physically, $\kappa_{clon}^{C}$ coincides with the maximum amount of work needed to generate $\Delta N$ copies of a pure state from the thermal state[25].

Since cloning is a special case of a deterministic classical computation, the bound (1) is attainable, and the minimum nonequilibrium cost of classical cloning is

$$c_{clon}^{C}(F) = \frac{\Delta N \, \Delta A}{kT \ln 2} + \log F. \tag{7}$$

This result generalises seminal results by Landauer and Bennett on the thermodynamics of classical cloning[7,54,55], extending them from the ideal scenario to realistic settings where the copying process is approximate. For systems with fully degenerate energy levels, one also has the equality $W_{clon}^{C}(F) = kT(\ln 2) c_{clon}^{C}(F)$, which provides the minimum amount of work needed to replicate classical information with a target level of accuracy.

## Nonequilibrium cost of quantum cloning

We now consider the task of approximately cloning quantum information[56]. The accuracy of quantum cloning is important both for foundational and practical reasons, as it is linked to the no signalling

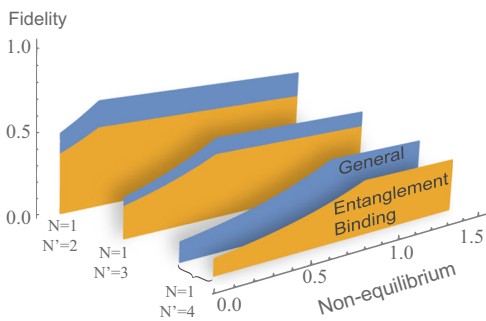

**Fig. 3 | Entanglement binding machines vs. general quantum machines.** The figure illustrates the accessible regions for the cloning fidelity when generating $N' = 2,3,4$ output copies from $N = 1$ input copy, in the case of qubits with degenerate Hamiltonian. The values of the fidelity in the blue region are attainable by general quantum machines, while the values in the orange region are attainable by entanglement binding machines. The difference between the two regions indicates a thermodynamic advantage of general quantum machines over all classical machines.

principle[57], to quantum cryptography[56], quantum metrology[58], and a variety of other quantum information tasks[59].

Here, we consider arbitrary cloning tasks where the set of single-copy states includes all energy eigenstates. This includes in particular the task of universal quantum cloning[60–62], where the input states are arbitrary pure states. The reverse entropy of universal quantum cloning is at least as large as the reverse entropy of classical cloning: the bound $\kappa_{\mathrm{clon}}^{\mathrm{Q}} \geq \kappa_{\mathrm{clon}}^{\mathrm{C}}$ follows immediately from Eq. (3), by restricting the optimisation to probability distributions that are concentrated on the eigenstates of the energy.

In Supplementary Note 5, we show that (i) the bound (1) is attainable for universal quantum cloning, and (ii) $\kappa_{\mathrm{clon}}^{\mathrm{Q}} = \kappa_{\mathrm{clon}}^{\mathrm{C}}$. These results imply that classical and quantum cloning exhibit exactly the same tradeoff between accuracy and nonequilibrium: for every value of the accuracy, the minimum nonequilibrium cost of information replication is given by Eq. (7) both in the classical and in the quantum case. An illustration of this fact is provided in Fig. 2. In terms of accuracy/nonequilibrium tradeoff, the only difference between classical and quantum cloning is that the classical tradeoff curve goes all the way up to unit fidelity, while the quantum tradeoff curve stops at a maximum fidelity, which is strictly smaller than 1 due to the no-cloning theorem[9,10].

Considering the differences between quantum and classical cloning, the fact that these two tasks share the same tradeoff curve is quite striking. An insight into this phenomenon comes from connection between the nonequilibrium cost and the time-reversed task of cloning. For fully degenerate systems, the time-reversed task is to transform $N'$ copies of a state into $N \leq N'$ copies of the same state, and in both cases it can be realised by discarding $N' - N$ systems. The reverse accuracy of this task is the same for both classical and quantum systems, and so is the reverse entropy. In the nondegenerate case, the analysis is more complex, but the conclusion remains the same.

Although classical and quantum cloning share the same tradeoff curve, in the following we will show that they exhibit a fundamental difference in the way the tradeoff is achieved: to achieve the fundamental limit, cloning machines must use genuinely quantum strategies.

## Limit on the accuracy of classical machines

Classical copy machines scan the input copies and produce replicas based on this information. Similarly, a classical machine for a general task can be modelled as a machine that measures the input and produces an output based on the measurement result. When this

approach is used at the quantum scale, it leads to a special class of quantum machines, known as entanglement breaking[63].

Here, we show that entanglement breaking machines satisfy a stricter bound. In fact, this stricter bound applies not only to entanglement breaking machines, but also to a broader class of machines, called entanglement binding[64]. An entanglement binding channel is a quantum channel that degrades every entangled state to a bound (a.k.a. PPT) entangled state[65,66]. In Methods, we show that the minimum nonequilibrium cost over all entanglement binding machines, denoted by $c_{\mathcal{T}}^{\mathrm{eb}}(F)$, must satisfy the inequality

$$c_{\mathcal{T}}^{\mathrm{eb}}(F) \geq \max\{\kappa_{\mathcal{T}}, \kappa_{\mathcal{T}^*}\} + \log F, \tag{8}$$

where $\kappa_{\mathcal{T}}$ is the reverse entropy of the state transformation task $\rho_x \to \rho'_x$, and $\kappa_{\mathcal{T}^*}$ is the reverse entropy of the transposed task $\mathcal{T}^*$, corresponding to the state transformation $\rho_x \mapsto (\rho'_x)^T$. This bound can be used to demonstrate that a thermodynamic advantage of general quantum machines over all entanglement binding machines, including in particular all classical machines.

## Quantum advantage in cloning

For quantum cloning, it turns out that no entanglement binding machine can achieve the optimal accuracy/nonequilibrium tradeoff. The reason for this is that the reverse entropy of the transpose task is strictly larger than the reverse entropy of the direct task, namely $\kappa_{\mathrm{clon}^*} > \kappa_{\mathrm{clon}}$. In Supplementary Note 6, we prove the inequality

$$\kappa_{\mathrm{clon}^*} \geq \kappa_{\mathrm{clon}} + \log \frac{d_{N+N'}}{d_{N'}} \frac{e^{-\frac{N'\Delta E}{kT}}}{1}, \tag{9}$$

where $\Delta E$ is the difference between the maximum and minimum energy, and, $d_K = (K + d - 1)!/[K!(d - 1)!]$ for $K = N$ or $K = N + N'$. Inserting this inequality into Eq. (8), we conclude that every entanglement binding machine necessarily requires a larger number of clean qubits compared to the optimal quantum machine.

When the energy levels are fully degenerate, we show that the bounds (8) and (9) are exact equalities. With this result at hand, we can compare the exact performance of entanglement binding machines and general quantum machines, showing that the latter achieve a higher accuracy for every given amount of nonequilibrium resources. The comparison is presented in Fig. 3.

Our result shows that entanglement binding machines are thermodynamically inefficient for the task of information replication. Achieving the ultimate efficiency limit requires machines that are able to preserve free (i.e., non-bound) entanglement. This observation fits with the known fact that classical machines cannot achieve the maximum copying accuracy allowed by quantum mechanics[61,62,67]. Here, we have shown that not only classical machines are limited in their accuracy, but also that, to achieve such limited accuracy, they require a higher amount of nonequilibrium resources. Interestingly, the thermodynamic advantage of general quantum machines vanishes in the asymptotic limit $N' \to \infty$, in which the optimal quantum cloning can be reproduced by state estimation[68–70].

## Thermodynamic benchmark for quantum memories and quantum communication

Quantum machines that preserve free entanglement also offer an advantage in the storage and transmission of quantum states, corresponding to the ideal state transformation $\rho_x \mapsto \rho_x$ where $x$ parametrises the states of interest. In theory, a noiseless quantum machine can achieve perfect accuracy at zero work cost. In practice, however, the transmission is always subject to errors and inefficiencies, resulting into nonunit fidelity and/or nonzero work. For this reason, realistic experiments that aim to demonstrate genuine quantum transmission or storage need criteria to demonstrate superior performance with

respect to all classical setups. A popular approach is to demonstrate an experimental fidelity larger than the maximum fidelity achievable by classical schemes[71–73]. In the qubit case, the maximum classical fidelity is $F_{max}^{eb} = 2/3$[74], and is often used as a benchmark for quantum communication experiments[75–77]. Here, we provide a different benchmark, in terms of the nonequilibrium cost needed to achieve a target fidelity $F$. In Supplementary Note 7, we show that the minimum nonequilibrium cost over all entanglement binding machines for the storage/transmission of qubit states is

$$c_{store/transmit}^{eb}(F) = \log\left[F + e^{\frac{\Delta E}{kT}}\frac{(2F-1)^2}{1-F}\right], \qquad (10)$$

Eq. (10) is valid for every qubit Hamiltonian and for every value of $F$ in the interval $[F_{min}^{eb}, F_{max}^{eb}]$, with $F_{max}^{eb} = 2/3$ and $F_{min}^{eb} = (e^{\frac{\Delta E}{kT}}+1)/(2e^{\frac{\Delta E}{kT}}+1)$. The minimum cost $c_{store/store}^{eb}(F)$ can be achieved by state estimation, and therefore can be regarded as the classical limit on the nonequilibrium cost.

For every $F > F_{min}$, the minimum nonequilibrium cost (10) is strictly larger than zero for every nondegenerate Hamiltonian. Since the nonequilibrium cost is a lower bound to the work cost, Eq. (10) implies that every entanglement binding machine with fidelity $F$ requires at least $kT(\ln 2) c_{store/transmit}^{eb}(F)$ work. This value can be used as a benchmark to certify genuine quantum information processing: every realistic setup that achieves fidelity $F$ with less than $kT \ln\left[F + e^{\frac{\Delta E}{kT}}(2F-1)^2/(1-F)\right]$ work will necessarily exhibit a performance that cannot be achieved by any classical setup. Notably, the presence of a thermodynamic constraint (either on the nonequilibrium or on the work) provides a way to certify a quantum advantage even for noisy implementations of quantum memories and quantum communication systems with fidelity below the classical fidelity threshold $F_{max} = 2/3$. A generalisation of these results for higher dimensional systems is provided in Supplementary Note 7.

## Discussion

An important feature of our bound (1) is that it applies also to state transformations that are forbidden by quantum mechanics, such as ideal quantum cloning or ideal quantum transposition. For state transformations that can be exactly implemented, instead, it is interesting to compare our bound with related results in the literature.

For exact implementations, the choice of accuracy measure is less important, and one can use any measure for which Eq. (1) yields a useful bound on the work cost. For example, consider the problem of generating a state $\rho$ from the equilibrium state. By choosing a suitable measure of accuracy (see Methods for the details), we find that the nonequilibrium cost for the state transition $\Gamma \mapsto \rho$ is equal to $D_{max}(\rho \parallel \Gamma)$, where $D_{max}(\rho \parallel \sigma) := \lim_{\alpha \to \infty} D_\alpha(\rho \parallel \sigma)$ is the max relative entropy, $D(\rho \parallel \sigma) := \log \text{Tr}[\rho^\alpha \sigma^{1-\alpha}]/(\alpha-1)$, $\alpha \geq 0$ being the the Rényi relative entropies. In this case, the nonequilibrium cost coincides (up to a proportionality constant $kT \ln 2$) with the minimal amount of work needed to generate the state $\rho$ without errors[25]. Similarly, one can consider the task of extracting work from the state $\rho$, corresponding to the state transition $\rho \mapsto \Gamma$. Ref. 25 showed that the maximum extractable work is $D_{min}(\langle\rho\rangle \parallel \Gamma) kT \ln 2$, where $D_{min}(\rho \parallel \sigma) := D_0(\rho \parallel \sigma)$ is the min relative entropy as per Datta's definition[78] and $\langle\rho\rangle$ is the time-average of $\rho$. This value can also be retrieved from our bound with a suitable choice of accuracy measure (see Supplementary Note 8 for the details). Smooth versions of these entropic quantities naturally arise by smoothing the task, that is, by considering small deviation from the input/output states that specify the desired state transformation (see Methods).

Our bound can also be applied to the task of information erasure with the assistance of a quantum memory[33]. There, a machine has access to a system $S$ and to a quantum memory $Q$, and the goal is to reset system $S$ to a pure state $\eta_S$, without altering the local state of the memory. When the initial states of system $SQ$ are drawn from a time-invariant subspace, our bound (1) implies that the work cost satisfies the inequality $W/(kT \ln 2) \geq D_{max}(\eta_S \otimes \gamma_Q \parallel \Gamma_{SQ}) - D_{max}(\tilde{\Gamma}_{SQ} \parallel \Gamma_{SQ})$, where $\tilde{\Gamma}_{SQ}$ is the quantum state obtained by projecting the Gibbs state onto the input subspace, and $\gamma_Q = \text{Tr}_S[\tilde{\Gamma}_{SQ}]$ is the marginal state of the memory. The bound is tight, and, for degenerate Hamiltonians, it matches the upper bound from ref. 33 up to logarithmic corrections in the error parameters (see Supplementary Note 9).

Another interesting issue is to determine when a given state transition $\rho \mapsto \rho'$ can be implemented without investing work. For states that are diagonal in the energy basis, a necessary and sufficient condition was derived in ref. 26, adopting a framework where catalysts are allowed. In this setting, ref. 26 showed that the state transition $\rho \mapsto \rho'$ can be implemented catalytically without work cost if and only if

$$D_\alpha(\rho' \parallel \Gamma_B) \leq D_\alpha(\rho \parallel \Gamma_A) \qquad \forall \alpha \geq 0. \qquad (11)$$

These conditions can be compared with our bound (1). In Methods, we show that, with a suitable choice of figure of merit, Eq. (1) implies the lower bound $W/(kT \ln 2) \geq D_{max}(\rho' \parallel \Gamma_B) - D_{max}(\rho \parallel \Gamma_A)$ for the perfect execution of the state transition $\rho \mapsto \rho'$. Hence, the work cost for the state transition $\rho \mapsto \rho'$ satisfies the bound $W/(kT \ln 2) \geq D_{max}(\rho' \parallel \Gamma_B) - D_{max}(\rho \parallel \Gamma_A)$, and the r.h.s. is nonpositive only if $D_{max}(\rho' \parallel \Gamma_B) \leq D_{max}(\rho \parallel \Gamma_A)$. The last condition is a special case of Eq. (11), corresponding to $\alpha \to \infty$. Notably, this condition and Eq. (11) are equivalent when the input and output states have well-defined energy, including in particular the case where the Hamiltonians of systems $A$ and $B$ are fully degenerate. Further discussion on the relation between quantum relative entropies and the cost of accuracy is provided in Supplementary Note 10.

While the applications discussed in the paper focussed on one-shot tasks, our results also apply to the asymptotic scenario where the task is to implement the transformation $\rho_x^{\otimes n} \mapsto \rho_x'^{\otimes n}$ in the large $n$ limit. In Methods we consider the amount of nonequilibrium per copy required by this transformation, allowing for small deviations in the input and output states. This setting leads to the definition of a smooth reverse entropy of a task, whose value per copy is denoted by $\kappa_{T,iid}$ and is shown to satisfy the bound

$$\kappa_{T,iid} \geq \max_x S(\rho' \parallel \Gamma_B) - S(\rho_x \parallel \Gamma_A), \qquad (12)$$

where $S(\rho \parallel \sigma) := \text{Tr}[\rho(\log \rho - \log \sigma)]$ is the quantum relative entropy.

In the special case where the state transformation $\rho_x \to \rho_x'$ can be implemented perfectly, and where $(\rho_x)_{x \in X}$ is the set of all possible quantum states of the input system, the r.h.s. of Eq. (12) (times $kT \ln 2$) coincides with the thermodynamic capacity introduced by Faist, Berta, and Brandão in ref. 79. In this setting, the results of ref. 79 imply that the thermodynamic capacity coincides with the amount of work per copy needed to implement the transformation $\rho_x \to \rho_x'$. Since the amount of work cannot be smaller than the nonequilibrium cost, this result implies that our fundamental accuracy/nonequilibrium tradeoff is asymptotically achievable for all information-processing tasks allowed by quantum mechanics.

In a different setting and with different techniques, questions related to the thermodynamical cost of physical processes have been studied in the field of stochastic thermodynamics[36]. Most of the works in this area focus on the properties of nonequilibrium steady states of classical systems with Markovian dynamics. An important result is a tradeoff relation between the relative standard deviation of the outputs associated with the currents in the nonequilibrium steady state and the overall entropy production[37–40]. This relation, called a thermodynamic uncertainty relation, is often interpreted as a tradeoff between the precision of a process and its thermodynamical cost. A difference with our work is that the notion of precision used in stochastic thermodynamics is not directly related to general

information-processing tasks. Another difference is that thermo-dynamic uncertainty relations do not always hold for systems outside the nonequilibrium steady state[39], whereas our accuracy/none-quilibrium tradeoff applies universally to all quantum systems. An interesting avenue of future research is to integrate the information-theoretic methods developed in this paper with those of stochastic thermodynamics, seeking for concrete physical models that approach the ultimate efficiency limits.

## Methods
### General performance tests
The performance of a machine in a given information-processing task can be operationally quantified by the probability to pass a test[73,80,81]. In the one-shot scenario, a general test $\mathcal{T}$ consists in preparing states of a composite system $AR$, consisting of the input of the machine and an additional reference system. The machine is requested to act locally on system $A$, while the reference system undergoes the identity process, or some other (generally noisy) process $\mathcal{R}_x$ implemented by the party that performs the test. Finally, the reference system and the output of the machine undergo a joint measurement, described by a suitable observable. The measurement outcomes are regarded as the score assigned to the machine. The test $\mathcal{T}$ is then described by the possible triples $(\rho_x, \mathcal{R}_x, O_x)_{x \in \mathsf{X}}$, consisting of an input state, a process on the reference system, and an output observable. In the worst case over all possible triples, one gets the accuracy $\mathcal{F}_\mathcal{T}(\mathcal{M}) := \min_x \mathrm{Tr}[O_x(\mathcal{M} \otimes \mathcal{R}_x)(\rho_x)]$, where $\mathcal{M}$ is the map describing the machine's action. Note that the dependence of the state $\rho_x$, transformation $\mathcal{R}_x$, and measurement $O_x$ can be arbitrary, and that the parameter $x$ can also be a vector $x = (x_1, \ldots, x_n)$. For example, the input state $\rho_x$ could depend only on the subset of the entries of the vector $x$, while the output observable $O_x$ could depend on all the entries, thus describing the situation where multiple observables are tested for the same input state.

Performance tests provide a more general way to define information-processing tasks. Rather than specifying a desired state transformation $\rho_x \mapsto \rho'_x$, one can directly specify a test that assigns a score to the machine. The test can be expressed in a compact way in the Choi representation[82]. In this representation, the test is described by a set of operators $(\Omega_x)_{x \in \mathsf{X}}$, called the performance operators[81], acting on the product of the input and output Hilbert spaces. The accuracy of the test has the simple expression $\mathcal{F}_\mathcal{T}(\mathcal{M}) = \min_x \mathrm{Tr}[M \Omega_x]$, where $M := (\mathcal{I}_A \otimes \mathcal{M})(|I_A\rangle\langle I_A|)$, $|I_A\rangle := \sum_i |i\rangle \otimes |i\rangle$ is the Choi operator of channel $\mathcal{M}$, and $\mathcal{I}_A$ is the identity on system $A$. In the following we will take each operator $\Omega_x$ to be positive semidefinite without loss of generality.

### Exact expression for the nonequilibrium cost
In Supplementary Note 1, we show that the nonequilibrium cost of a general task $\mathcal{T}$ can be evaluated with the expression $c_\mathcal{T}(F) = \max_\mathbf{p} c_{\mathcal{T},\mathbf{p}}(F)$, where the minimum is over all probability distributions $\mathbf{p} = (p_x)_{x \in \mathsf{X}}$ and

$$c_{\mathcal{T},\mathbf{p}}(F) = \log \max_{\substack{X_A \otimes I_B + z\,\Omega_\mathbf{p} \le \Gamma' \otimes Y_B \\ Tr[\Gamma_B Y_B] \le 1}} \mathrm{Tr}[X_A] + zF, \qquad (13)$$

with $\Omega_\mathbf{p} := \sum_x p_x \Omega_x$, $\Gamma' := \Pi_A \Gamma_A \Pi_A$. Here, the maximisation runs over all Hermitian operators $X_A(Y_B)$ acting on system $A$ ($B$) and over all real numbers $z$.

For every fixed probability distribution $\mathbf{p}$, the evaluation of $c_{\mathcal{T},\mathbf{p}}(F)$ is a semidefinite programme[83], and can be solved numerically for low-dimensional systems. A simpler optimisation problem arises by setting

$X_A = 0$, which provides the lower bound

$$c_{\mathcal{T},\mathbf{p}}(F) \ge \log \max_{\substack{z\,\Omega_\mathbf{p} \le \Gamma' \otimes Y_B \\ Tr[\Gamma_B Y_B] \le 1}} zF$$
$$= H(A|B)_{\omega_{\mathcal{T},\mathbf{p}}} + \log F. \qquad (14)$$

(see Supplementary Note 1 for the derivation).

### Time-reversed tasks and reverse entropy
For a given task $\mathcal{T}$, implemented by operations with input $A$ and output $B$, we define a time-reversed task $\mathcal{T}^{\mathrm{rev}}$, implemented by operations with input $B$ and output $A$. For example, consider the case where the direct task is to transform pure states into pure states, according to a given mapping $\rho_x \mapsto \rho'_x$, on a quantum system with fully degenerate energy levels, and the accuracy of the implementation measured by the fidelity $\mathcal{F}_\mathcal{T}(\mathcal{M}) = \min_x \mathrm{Tr}[\rho'_x \mathcal{M}(\rho_x)]$. In this case, the time-reversed task is to implement the transformation $\rho'_x \mapsto \rho_x$, using some channel $\mathcal{M}$ with input $B$ and output $A$. The accuracy is then given by the reverse fidelity $\mathcal{F}_{\mathcal{T}_{\mathrm{rev}}}(\mathcal{M}) = \min_x \mathrm{Tr}[\rho_x \mathcal{M}(\rho'_x)]$. More generally, we define the time-reversed task $\mathcal{T}_{\mathrm{rev}}$ in terms of a time reversal for quantum operations[41,42], related to Petz's recovery map[43–45]. The specific version of the time reversal used here maps the states $\rho_x$ into the observables $\widetilde{O}_x := \Gamma_A^{-\frac{1}{2}} \rho_x \Gamma_A^{-\frac{1}{2}}$ and the observables $O_x$ into the (unnormalised) states $\widetilde{\rho}_x := \Gamma_B^{\frac{1}{2}} O_x \Gamma_B^{\frac{1}{2}}$[42]. The reverse accuracy then becomes $\mathcal{F}_{\mathcal{T}_{\mathrm{rev}}}(\mathcal{M}) := \min_x \mathrm{Tr}[\widetilde{O}_x \mathcal{M}(\widetilde{\rho}_x)]$.

For a general information-processing task with performance operators $(\Omega_x)_{x \in \mathsf{X}}$, we define the time-reversed task $\mathcal{T}_{\mathrm{rev}}$ with performance operators $(\Omega_x^{\mathrm{rev}})_{x \in \mathsf{X}}$ defined by

$$\Omega_x^{\mathrm{rev}} := (\Gamma_B^{1/2} \otimes \Gamma_A^{-1/2}) E_{AB} \Omega_x^T E_{AB} (\Gamma_B^{1/2} \otimes \Gamma_A^{-1/2}), \qquad (15)$$

where $E_{AB} : \mathcal{H}_A \otimes \mathcal{H}_B \to \mathcal{H}_B \otimes \mathcal{H}_A$ is the unitary operator that exchanges systems $A$ and $B$. The reverse accuracy of a generic quantum channel $\mathcal{M}$ is then given by $\mathcal{F}_{\mathcal{T}_{\mathrm{rev}}}(\mathcal{M}) := \min_x \mathrm{Tr}[\Omega_x^{\mathrm{rev}} M]$, where $M$ is the Choi operator of $\mathcal{M}$. The maximum of the reverse accuracy over all quantum channels can be equivalently expressed in terms of a conditional min-entropy: indeed, one has

$$F_{\mathcal{T}_{\mathrm{rev}}}^{\max} = \max_\mathcal{M} \mathcal{F}_{\mathcal{T}_{\mathrm{rev}}}(\mathcal{M})$$
$$= \max_{M:M \ge 0, \mathrm{Tr}_A[M] = I_B} \min_x \mathrm{Tr}[\Omega_x^{\mathrm{rev}} M] \qquad (16)$$
$$= \max_{M:M \ge 0, \mathrm{Tr}_A[M] = I_B} \min_\mathbf{p} \mathrm{Tr}[\omega_{\mathcal{T},\mathbf{p}} M]$$

where the minimum is over all probability distributions $\mathbf{p} = (p_x)$, and $\omega_{\mathcal{T},\mathbf{p}} := (\sum_x p_x \Omega_x^{\mathrm{rev}})^T$. Using von Neumann's minimax theorem, we then obtain

$$F_{\max}^{\mathcal{T}_{\mathrm{rev}}} = \min_\mathbf{p} \max_{M:M \ge 0, \mathrm{Tr}_A[M] = I_B} \mathrm{Tr}[\omega_{\mathcal{T},\mathbf{p}} M]$$
$$= \min_\mathbf{p} 2^{-H_{\min}(A|B)_{\omega_{\mathcal{T},\mathbf{p}}}} \qquad (17)$$
$$= 2^{-\max_\mathbf{p} H_{\min}(A|B)_{\omega_{\mathcal{T},\mathbf{p}}}},$$

where the second equality follows from the operational interpretation of the min-entropy[45]. Taking the logarithm on both sides of the equality, we then obtain the relation $\kappa_\mathcal{T} := -\log F_{\max}^{\mathcal{T}_{\mathrm{rev}}} = \max_\mathbf{p} H_{\min}(A|B)_{\omega_\mathbf{p}}$, corresponding to Eq. (3) in the main text. The bound (1) then follows from the relation $c_\mathcal{T}(F) = \max_\mathbf{p} c_{\mathcal{T},\mathbf{p}}$ and from Eq. (14).

### Bounds on the reverse entropy
When the test $\mathcal{T}$ consists in the preparation of a set of states $(\rho_x)_{x \in \mathsf{X}}$ of system $A$ and in the measurement of a set of observables $(O_x)_{x \in \mathsf{X}}$ on

system $B$, the reverse entropy can be lower bounded as

$$\kappa_{\mathcal{T}} \geq \max_x \ -\log \mathrm{Tr}[\Gamma_B \, O_x] - D_{\max}(\rho_x \parallel \Gamma_A), \qquad (18)$$

with the equality holding when $|X| = 1$ (see Supplementary Note 10 for the proof and for a discussion on the relation between the nonequilibrium cost of a state transformation task $\rho_x \mapsto \rho'_x, \forall x \in X$ and the nonequilibrium cost of the individual state transitions $\rho_x \mapsto \rho'_x$ for a fixed value of $x$).

A possible choice of observable is $O_x = P_x$, where $P_x$ is the projector on the support of the target state $\rho'_x$. In this case, the bound (18) becomes

$$\kappa_{\mathcal{T}} \geq \max_x \ D_{\min}(\rho'_x \parallel \Gamma_B) - D_{\max}(\rho_x \parallel \Gamma_A). \qquad (19)$$

An alternative choice of observables is $O_x = \Gamma^{-1/2} |\psi_x\rangle\langle\psi_x| \Gamma^{-1/2} / \parallel \Gamma^{-1/2} \rho'_x \Gamma^{-1/2} \parallel$, where $|\psi_x\rangle$ is the normalised eigenvector corresponding to the maximum eigenvalue of $\Gamma_B^{-1/2} \rho'_x \Gamma_B^{-1/2}$. With this choice, the bound (18) becomes $\kappa \geq \max_x D_{\max}(\rho'_x \parallel \Gamma_B) - D_{\max}(\rho_x \parallel \Gamma_A)$, with the equality when $|X| = 1$. Combining this bound with Eq. (1), we obtain the following

**Proposition 1.** If there exists a quantum channel $\mathcal{M}$ such that $\mathcal{M}(\rho_x) = \rho'_x$ for every $x \in X$, then its nonequilibrium cost satisfies the bound $c(\mathcal{M}) \geq \max_x D_{\max}(\rho'_x \parallel \Gamma_B) - D_{\max}(\rho_x \parallel \Gamma_A)$.

The proposition follows from Eq. (1) and from the fact that the channel $\mathcal{M}$ has accuracy $\mathcal{F}(\mathcal{M}) = \min_x \mathrm{Tr}[\mathcal{M}(\rho_x) O_x] = 1$.

### Smooth reverse entropy
For an information-processing task $\mathcal{T}$ with operators $(\Omega_x)_{x \in X}$, one can consider an approximate version, described by another task $\mathcal{T}'$ with operators $(\Omega'_x)_{x \in X'}$ that are close to $(\Omega_x)_{x \in X}$ with respect to a suitable notion of distance. One can then define the worst (best) case smooth reverse entropy of the task $\kappa_{\mathcal{T}, \epsilon}$ as the maximum (minimum) of $\kappa_{\mathcal{T}'}$ over all tasks $\mathcal{T}'$ that are within distance $\epsilon$ from the given task. The choice between the worst case and the best case irreversibility depends on the problem at hand. A best case irreversibility corresponds to introducing an error tolerance in the task, thus discarding low-probability events that would result in a higher cost[25,33]. Instead, a worst case irreversibility can be used to model noisy scenarios, where the input states may not be the ones in the ideal information-processing task. An example of this situation is the experimental implementation of quantum cloning, where the input states may not be exactly pure.

Smoothing is particularly useful in the asymptotic scenario. Consider the test $\mathcal{T}_n$ that consists in preparing a multi-copy input state $\rho_x^{\otimes n}$ and measuring the observable $P_{x,n}$, where $P_{x,n}$ is the projector on the support of the target state $\rho_x'^{\otimes n}$. A natural approximation is to allow, for every $x \in X$, all inputs $\rho_{y,n}$ that are $\epsilon$-close to $\rho_x^{\otimes n}$, and all outputs $\rho'_x$ that are $\epsilon$-close to $\rho_x'^{\otimes n}$. Choosing $\kappa_{\mathcal{T}_n, \epsilon}$ to be the worst case smooth reverse entropy of the task $\mathcal{T}_n$, Eq. (19) gives the bound $\kappa_{\mathcal{T}_n, \epsilon} \geq \max_x D_{\min}^\epsilon(\rho_x'^{\otimes n} \parallel \Gamma_B^{\otimes n}) - D_{\max}^\epsilon(\rho_x^{\otimes n} \parallel \Gamma_A^{\otimes n})$, where $D_{\min}^\epsilon$ and $D_{\max}^\epsilon$ are the smooth versions of $D_{\min}$ and $D_{\max}$[78]. One can then define the regularised reverse entropy of the task as $\kappa_{\mathcal{T}, iid} := \lim_{\epsilon \to 0} \sup_n \kappa_{\mathcal{T}_n, \epsilon}/n$. Using the relations $\lim_{\epsilon \to 0} \sup_n D_{\min}^\epsilon(\rho_x'^{\otimes n} \parallel \Gamma_B^{\otimes n})/n = S(\rho'_x \parallel \Gamma_B)$ and $\lim_{\epsilon \to 0} \inf_n D_{\max}^\epsilon(\rho_x^{\otimes n} \parallel \Gamma_A^{\otimes n})/n = S(\rho_x \parallel \Gamma_A)$[78] we finally obtain the bound $\kappa_{\mathbf{T}, iid} \geq \max_x S(\rho'_x \parallel \Gamma_B) - S(\rho_x \parallel \Gamma_A)$. The quantity on the r.h.s. coincides with the thermodynamic capacity introduced by Faist, Berta, and Brandão in ref. [79], where it was shown that the thermodynamic capacity coincides with the amount of work per copy needed to implement the transformation $\rho_x \to \rho'_x$. Combining this result with our bounds, we obtain that the fundamental accuracy/nonequilibrium

in Eq. (1) is asymptotically achievable for all transformations allowed by quantum mechanics.

### Limit for entanglement binding channels
Entanglement binding channels generally satisfy a more stringent limit than (1). The derivation of this strengthened limit is as follows: first, the definition of an entanglement binding channel $\mathcal{P}$ implies that the map $\mathcal{P}^{PT}$ defined by $\mathcal{P}^{PT}(\rho) := [\mathcal{P}(\rho)]^T$ is a valid quantum channel. Now, the nonequilibrium cost of the channels $\mathcal{P}$ and $\mathcal{P}^{PT}$ is given by $D_{\max}(\mathcal{P}(\Pi_A \Gamma_A \Pi_A \parallel \Gamma_B))$ and $D_{\max}(\mathcal{P}^{PT}(\Pi_A \Gamma_A \Pi_A \parallel \Gamma_B))$ (cf. Supplementary Note 1). Since the max relative entropy satisfies the relation $D_{\max}(\rho \parallel \sigma) = D_{\max}(\rho^T \parallel \sigma^T)$ for every pair of states $\rho$ and $\sigma$, the costs of $\mathcal{P}$ and $\mathcal{P}^{PT}$ are equal.

The second step is to note that the accuracy of the channel $\mathcal{P}$ for the task specified by the performance operators $(\Omega_x)$ is equal to the accuracy of the channel $\mathcal{P}^{PT}$ for the task specified by the performance operators $(\Omega_x^{T_B})$, where $T_B$ denotes the partial transpose over system $B$. Applying the bound (1) to channel $\mathcal{P}^{PT}$, we then obtain the relation

$$\begin{aligned} c(\mathcal{P}) &= c(\mathcal{P}^{PT}) \\ &\geq \kappa_{\mathcal{T}^*} + \log F, \end{aligned} \qquad (20)$$

where $\kappa_{\mathcal{T}^*}$ is the reverse entropy of the transpose task $\mathcal{T}^*$, with performance operators $(\Omega_x^{T_B})$. Since entanglement binding channel is subject to both bounds (1) and (20), Eq. (8) holds.

## Data availability
The authors declare that the data supporting the findings of this study are available within the paper and in the supplementary information files.

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

## Acknowledgements

G.C. acknowledges a helpful discussion with Nilanjiana Datta on the quantum extensions of Rényi relative entropies. F.M. acknowledges Yuxiang Yang, Mile Gu, and Oscar Dahlsten for helpful comments that helped improving the presentation. This work was supported by the Hong Kong Research Grant Council through grants 17326616 (G.C.) and 17300918 (G.C.), and through the Senior Research Fellowship Scheme via SRFS2021-7S02 (G.C.), by the Swiss National Science Foundation via grant 200021_188541 (R.R.), by the National Natural Science Foundation of China through grants 11675136 (G.C.), 11875160 (M.Y.) and U1801661 (M.Y.), by the Key R&D Programme of Guangdong province through grant 2018B030326001 (M.Y.), by the Guangdong Provincial Key Laboratory through grant c1933200003 (M.Y.), the Guangdong Innovative and Entrepreneurial Research Team Programme via grant 2016ZT06D348 (M.Y.), the Science, Technology and Innovation Commission of Shenzhen Municipality through grant KYTDPT20181011104202253 (M.Y.). Research at the Perimeter Institute is supported by the Government of Canada through the Department of Innovation, Science and Economic Development Canada and by the Province of Ontario through the Ministry of Research, Innovation and Science.

## Author contributions

G.C. and M.Y. proposed the initial idea. G.C. introduced the notion of reverse entropy and proved the bound on the nonequilibrium cost. F.M. derived the achievability condition and computed the nonequilibrium cost of quantum cloning. R.R. and G.C. devised the notion of smoothing for the reverse entropy, and developed the connections with information erasure, work cost, and work extraction. All authors contributed substantially to the development of the preparation of the paper. G.C. and F.M. contributed equally.

## Competing interests

The authors declare no competing interests.
