## [Peer Review File · Nature Communications]

The nonequilibrium cost of accurate information processingREVIEWER COMMENTS

Reviewer #1 (Remarks to the Author):

This manuscript considers the question of what is the minimal cost of non-equilibrium sources required for transforming a given set of inputs to a given set of outputs with a prescribed level of accuracy. It is shown that this problem can, when suitably formulated, be cast as a semidefinite program. The nature of the construction, in terms of tests of the output states, allows not only the analysis of physically implementable operations, but also operations that are impossible to implement exactly. In this sense, the authors compare classical and quantum replication. For 'physically allowed' operations, the authors moreover show that their results can be used to regain several of previous results in the literature, such as a bound on the work extraction from non-equilibrium states, as well as the cost of erasure with quantum side information.

In my view, this manuscript provides a very relevant contribution to the thermodynamics of information processing, that should be of interest to the resource theory and quantum thermodynamics community. Personally, I think that the focus on the accuracy and performance tests, rather than on enforcing particular operations, is an interesting and fruitful idea. All in all, I would say that results fits well within the scope of Nature Communications, and I recommend publication.

Overall, I think that the paper already is in a rather good shape, and could be published in its current form. I have nevertheless collected a few remarks that the authors may want to consider.

* In my view, the presentation of the background of the problem is good, and the collection of references are reasonable. However, I might point out that in the context of stochastic thermodynamics, there has recently been a quite vigorous development that considers the thermodynamic cost of accuracy of processes (e.g., the degree of systematic and predictable motion in molecular motors). One, rather arbitrary, example of this literature is ["Stochastic thermodynamics: From principles to the cost of precision", Udo Seifert, Physics A: Statistical Mechanics and its Applications, Vol. 504, p. 176 (2018).] The title alone suggests that it could be good to comment on the relation, or lack of relation, with the current manuscript. Let me emphasize that I do not claim that there is any overlap in actual results; the entire technical machinery and philosophy of the current manuscript is quite orthogonal. However, the lack of discussion might look odd from the point of view of the stochastic thermodynamics community. More interestingly, there might be connections worth exploring (although I have no illusions that this would be easy).

* p. 2, second column introduces the notion of nonequilibrium cost $c(M)$ of a channel M . I have got the impression that this concept already was introduced in Ref. [32]. Although [31,32] are cited above equation (1), the current formulation is somewhat blurry as to what concepts and findings are introduced in the current manuscript, and what are reiterations from [31,32].

* The first time that I read the paragraph "Definition of thermodynamic complexity" I found it rather confusing. I understood the intention more clearly first after reading paragraph "General performance tests" in the Methods section. If the reader should get a more immediate understanding upon a linear first read, I would suggest that these explanations require some more efforts. In the definition of ω_P , I got the impression that O_x in some sense captures the output (or rather the test of the output). Potentially, one could more explicitly point out the role of O_x , and refer back to the previous discussion about O_x and $F(M)$.

* Speaking of O_x and the accuracy $F(M)$, I believe that it could be good to clarify the underlying conceptual assumptions behind the construction of $F(M)$. (Not necessarily in the main text.) I get the impression that there is only one measurement setting O_x assumed for each possible state ρ_x .

Spontaneously, I would expect that a collection of measurement settings (with independent 'choices') for each state ρ_x would be the more general situation, and that a single measurement setting per state requires more assumptions and yields less power. In particular, I would guess that the single measurement setting requires that we trust the description of the measurement devices.

* When first reading the phrase "thermodynamic complexity", my spontaneous association was to computational complexity, and that there would be some direct association between these two concepts. However, as far as I understand, there is no such connection. Not that this is particularly crucial, but could not this choice of terminology risk to cause unnecessary confusion with current efforts to connect thermodynamics and computational complexity?

* In the results section, both the resource theory of Gibbs preserving operations, and the resource theory of thermal operations are discussed. From the main text alone I did unfortunately not get a clear understanding of the role, if any, of these two for the results of the paper.

* p. 3, below Theorem 1, it is said that this is proved in Supplementary Note 2, but is it not rather supplementary Note 1?

* In "Semidefinite program for the accuracy-nonequilibrium tradeoff" of the Supplemental Material, in relation to equation (8), I believe that it would be helpful to refer to, or briefly recall, the concepts of performance operators discussed in the paragraph "General performance tests" in the Methods section of the main text.

Reviewer #2 (Remarks to the Author): Review of “The nonequilibrium cost of accurate information processing”

Summary:

Meng et al. study the thermodynamics of general information processing tasks in quantum physics, defined by the relation between certain input and output states (rather than by specifying a channel). They quantify the accuracy of such a task via the expectation value of certain test observables. State transformations generally come with some thermodynamic cost – i.e., if one is restricted to using thermal processes (here, Gibbs-preserving maps), then extra resources are required, for instance in the form of an information battery or work battery. The authors identify the minimal cost $c(\mathcal{M})$ of a given channel \mathcal{M} (found in earlier work), then define c for a given information processing task as the smallest cost incurred by implementing the task with an accuracy above some threshold F . They also show how to compute this quantity with semidefinite programming.

The main result lower-bounds c in terms of just the “task” (rather than a specific channel). This uses the introduced “thermodynamic complexity” κ . The advantages of this bound are claimed to be its computability and the fact that it incorporates the parameter of accuracy. Therefore it can even be applied for tasks that are impossible to perform perfectly. The bound is not always attainable, but some sufficient conditions are given, such as classical computation. It is applied to the task of “replication” (imperfect cloning). The authors also prove that another bound exists specifically for entanglement binding machines (outputting no distillable entanglement), showing that such a machine must incur an additional cost for the replication task. Finally, they also demonstrate how previously-known bounds on work extraction and erasure costs can be rederived from the presented framework.

Main comments:

1. Can the previously introduced quantity c_ϵ (as defined in the supplementary notes) not also in principle be used for studying approximate instances of impossible tasks? The error ϵ for this is defined instead in terms of a distance measure. So would the problem just be that the resulting quantity would be more difficult to work with and compute? Or does the quantity c in the manuscript have a conceptual advantage as well?
2. On a related point: under eq 10, it seems the comparison between bounds in the case of the transpose task is not fair – eq 10 is for a set of state transformations, not a channel. Presumably one could apply eq 10 instead as follows: find a channel that implements the operation (say, transpose) to the desired accuracy, and then compute the α -Rényi relative entropies for the inputs and resulting outputs with that channel. Do the authors know how the resulting bound would compare with the one derived here?
3. I am uneasy with the terminology of “thermodynamic complexity”. Of course the word complexity has many different meanings across different fields, but it would be helpful if the quantity introduced here has at least some kind of connection with one of the established meanings. It is not clear to me why κ is much more than a measure of departure from equilibrium. As the authors say, in the classical case it is precisely that. The quantum case is more subtle, as shown by considering non-CP maps such as the transpose operation – but precisely what κ really measures beyond nonequilibrium is not explored very well. Could the authors give some more clarity on this and justify their terminology? Is there any relation with some recent notions of complexity relevant to thermodynamics, e.g. arXiv:2110.11371? (It appears not.)
4. Similarly, apart from giving a useful computable bound to the cost c , what thermodynamical insight do we get from considering κ ?
5. Is there any interesting relation with Ref. [30], which also characterises state transformations in terms of the min conditional entropy?
6. The writing is very good and should be accessible to a reasonable large readership. However the main text sometimes seems over-simplified to its detriment. For example, some quantities (including D_{\max} , H_{\min}) are never defined in the main text. For non-experts, definitions (perhaps left to the methods section) and some description of the meaning of the quantities, would be very helpful.
7. Statements about attainability of the bounds generally assume the ability to perform general Gibbs-preserving maps, rather than the better physically motivated subset of thermal operations. This issue is acknowledged by the authors; while it would be better to have a result better suited to thermal operations, this seems like a much harder problem.

Overall, I find that this manuscript addresses a well-motivated fundamental problem in the field of quantum thermodynamics, and is technically impressive, so certainly publishable. However, I do not see a strong conceptual advance over previous work. If I understand correctly, the definition of accuracy employed here (using observable expectation values rather than distance measures) results in a computationally more tractable version of the results from Refs. [31,32]. This appears to be the main contribution of the manuscript. While valuable, apart from this, it is not obvious what insight one can get from the introduced thermodynamic complexity. The examples studied are interesting but arguably not very useful since perfect cloning is anyhow an impossible task. In summary, I therefore do not see a convincing case for publication in this journal, unless the authors can provide some more remarkable insight or applications.

Minor comments:

1. Perhaps consider altering the notation to make it clear that the quantities c and κ depend on F ?
2. Above eq 1: it may help for clarity to also state that invariance under time evolution means the state is diagonal in the energy eigenbasis.
3. In the first paragraph of methods: “fixed” channel presumably means the identity?
4. Under eq 13 in methods: a bound is derived taking the observable as the projector onto the support of the output. When the output is mixed, $F = 1$ does not guarantee unit fidelity – does this show a downside of the approach used here?
5. At many points in the supplementary notes, H is written instead of H_{\min} .
6. “Cloning” and “replication” seem to be used interchangeably – please clarify. The latter appears to be non-standard terminology: does it simply refer to approximate cloning?

Response to Reviewers and List of Changes

We thank both Referees for their valuable comments, all of which have been addressed in the revised version of the manuscript. We are especially pleased with this new version, which includes also a number of new results, including in particular

- (1) A physical interpretation of our entropic quantity in terms of a time-reversal of the original information processing task
- (2) The exact tradeoff between work cost and accuracy for the task of information erasure.
- (3) A thermodynamic benchmark for quantum memories and quantum communication systems

The paper includes also several improvements in the presentation of the framework. The most significant changes are highlighted in blue for better readability.

A summary of the main changes is provided in the following:

- updated the abstract to include the new results
- included mention to stochastic thermodynamics in Introduction and Discussion
- Introduction updated to include the new results
- Improved presentation of the framework for the accuracy nonequilibrium tradeoff
- Introduced notion of time-reversed tasks
- Added an expanded discussion of the relation between nonequilibrium and work cost, including the exact expression of the work cost of the work cost of information erasure
- Added benchmark for storage and transmission of quantum data
- Improved discussion on quantum Rényi entropies and work cost of quantum processes (including new derivations in Supplementary Note 10)
- Added updated the Methods section to match the presentation in the main text.
- References to stochastic thermodynamics added.

Response to Reviewer #1

This manuscript considers the question of what is the minimal cost of non-equilibrium sources required for transforming a given set of inputs to a given set of outputs with a prescribed level of accuracy. It is shown that this problem can, when suitably formulated, be cast as a semidefinite program. The nature of the construction, in terms of tests of the output states, allows not only the analysis of physically implementable operations, but also operations that are impossible to implement exactly. In this sense, the authors compare classical and quantum replication. For 'physically allowed' operations, the authors moreover show that their results can be used to regain several of previous results in the literature, such as a bound on the work extraction from non-equilibrium states, as well as the cost of erasure with quantum side information.

In my view, this manuscript provides a very relevant contribution to the thermodynamics of information processing, that should be of interest to the resource theory and quantum thermodynamics community. Personally, I think that the focus on the accuracy and performance tests, rather than on enforcing particular operations, is an interesting and fruitful idea. All in all, I would say that results fits well within the scope of Nature Communications, and I recommend publication.

We thank the Referee for their accurate summary of our results and for the positive recommendation towards publication in Nature Communications. We especially appreciate that the Referee regards our work as a “*very relevant contribution to the thermodynamics of information processing,*” and finds the use of performance tests a fruitful approach. Exploring this idea was indeed our original motivation: we wished to find a direct relation between the accuracy of a task (operationally quantified by a performance test) and the thermodynamic resources needed to achieve it. This approach has been further highlighted in the revised version of the manuscript.

Overall, I think that the paper already is in a rather good shape, and could be published in its current form. I have nevertheless collected a few remarks that the authors may want to consider.

** In my view, the presentation of the background of the problem is good, and the collection of references are reasonable. However, I might point out that in the context of stochastic thermodynamics, there has recently been a quite vigorous development that considers the thermodynamic cost of accuracy of processes (e.g., the degree of systematic and predictable motion in molecular motors).*

One, rather arbitrary, example of this literature is [“Stochastic thermodynamics: From principles to the cost of precision”, Udo Seifert, Physics A: Statistical Mechanics and its Applications, Vol. 504, p. 176 (2018).] The title alone suggests that it could be good to comment on the relation, or lack of relation, with the current manuscript. Let me emphasize that I do not claim that there is any overlap in actual results; the entire technical machinery and philosophy of the current manuscript is quite orthogonal. However, the lack of discussion might look odd from the point of view of the stochastic thermodynamics community. More interestingly, there might be connections worth exploring (although I have no illusions that this would be easy).

Thank you for this pointing out these related developments. In the revised version, we mention them, first with a brief sentence in the Introduction, and then with a more detailed discussion at the end of the Discussion section. As you mention, the framework of stochastic thermodynamics is quite orthogonal to the resource-theoretic framework adopted in this paper, and the connections between them are rather indirect, although worth exploring. In the Discussion, we now discuss some of the results from stochastic thermodynamics, some of the differences with our results, and we highlight the exploration of the connections as an interesting direction of future research.

** The first time that I read the paragraph “Definition of thermodynamic complexity” I found it rather confusing. I understood the intention more clearly first after reading paragraph “General performance tests” in the Methods section. If the reader should get a more immediate understanding upon a linear first read, I would suggest that these explanations require some more efforts. In the definition of ω_P , I got the impression that O_x in some sense captures the output (or rather the test of the output). Potentially, one could more explicitly point out the role of O_x , and refer back to the previous discussion about O_x and $F(M)$.*

We appreciate that the definition of the “thermodynamic complexity” (now called differently in the revised version) was pretty abstract, and we thank the Referee for the helpful suggestions given above.

In the revised version, this part has substantially changed, due to new results that give a more direct interpretation of the “thermodynamic complexity.” To avoid potential ambiguities with the use of the term “complexity” in other fields, we opted for the term “reverse entropy” instead of “thermodynamic complexity.” The reverse entropy is now defined in term of a time reversal of the original task. In the time-reversed task, the role of the input states ρ_x and of the output observables O_x are exchanged. The reverse entropy is then defined as $-\log F_{\text{rev}}^{\text{max}}$ where $F_{\text{rev}}^{\text{max}}$ is the maximum accuracy allowed by quantum mechanics for the execution of the time-reversed task. We then proceed to show that this definition is equivalent to the one originally provided in the previous version of the manuscript.

** Speaking of O_x and the accuracy $F(M)$, I believe that it could be good to clarify the underlying conceptual assumptions behind the construction of $F(M)$. (Not necessarily in the main text.) I get the impression that there is only one measurement setting O_x assumed for each possible state ρ_x . Spontaneously, I would expect that a collection of measurement settings (with independent ‘choices’) for each state ρ_x would be the more general situation, and that a single measurement setting per state requires more assumptions and yields less power. In particular, I would guess that the single measurement setting requires that we trust the description of the measurement devices.*

Thank you for this valuable observation. In fact, the case of multiple observables for the same input state is included in our framework. Our parameter x can be general and, in particular, can be a vector with multiple components $x = (x_1, x_2, \dots, x_n)$. The dependence of the input state and of the output measurement on the parameter x can be arbitrary, and includes in particular the case where the input state ρ_x could depend only on a subset (x_1, \dots, x_k) of the entries of x , while the output measurement O_x could depend on all the components. In this way, our settings would include the scenario where multiple measurements are performed for the same input state. This point is now clarified in the main text.

** When first reading the phrase “thermodynamic complexity”, my spontaneous association was to computational complexity, and that there would be some direct association between these two concepts. However, as far as I understand, there is no such connection. Not that this is particularly crucial, but could not this choice of terminology risk to cause unnecessary confusion with current efforts to connect thermodynamics and computational complexity?*

We agree that there is indeed such a risk, and we changed the name of the quantity to “reverse entropy” in the revised version of the paper.

** In the results section, both the resource theory of Gibbs preserving operations, and the resource theory of thermal operations are discussed. From the main text alone I did unfortunately not get a clear understanding of the role, if any, of these two for the results of the paper.*

The distinction between Gibbs preserving and thermal operations plays a role in the definition of the quantities we consider. The “nonequilibrium cost” of a quantum process M is the number of clean qubits needed to realise M in a scheme using Gibbs preserving operations (as in Figure 1 of the paper). Instead, the “work cost” is (up to a factor $kT \ln 2$) the number of clean qubits needed to realise M in a similar scheme using thermal operations. Since the thermal operations are a (generally strict) subset of the Gibbs preserving operations, the nonequilibrium cost is a lower bound to the work cost. This point is now explained explicitly in the revised version, where we state the lower bound explicitly and briefly discuss its achievability in special cases, including the notable example of information erasure.

** p. 3, below Theorem 1, it is said that this is proved in Supplementary Note 2, but is it not rather supplementary Note 1?*

Thanks for spotting this typo, which has been corrected in the revised version.

** In “Semidefinite program for the accuracy-nonequilibrium tradeoff” of the Supplemental Material, in relation to equation (8), I believe that it would be helpful to refer to, or briefly recall, the concepts of performance operators discussed in the paragraph “General performance tests” in the Methods section of the main text.*

Thank you for the suggestion; the notion of performance test is now recalled before Eq. (8) in the Supplemental Material, including both reference to the Methods section and a brief summary of the main concepts.

Response to Reviewer #2

Summary. Meng et al. study the thermodynamics of general information processing tasks in quantum physics, defined by the relation between certain input and output states (rather than by specifying a channel). They quantify the accuracy of such a task via the expectation value of certain test observables. State transformations generally come with some thermodynamic cost – i.e., if one is restricted to using thermal processes (here, Gibbs-preserving maps), then extra resources are required, for instance in the form of an information battery or work battery. The authors identify the minimal cost $c(M)$ of a given channel M (found in earlier work), then define c for a given information processing task as the smallest cost incurred by implementing the task with an accuracy above some threshold F . They also show how to compute this quantity with semidefinite programming. The main result lower-bounds c in terms of just the “task” (rather than a specific channel). This uses the introduced “thermodynamic complexity” κ . The advantages of this bound are claimed to be its computability and the fact that it incorporates the parameter of accuracy. Therefore it can even be applied for tasks that are impossible to perform perfectly. The bound is not always attainable, but some sufficient conditions are given, such as classical computation.

It is applied to the task of “replication” (imperfect cloning). The authors also prove that another bound exists specifically for entanglement binding machines (outputting no distillable entanglement), showing that such a machine must incur an additional cost for the replication task. Finally, they also demonstrate how previously-known bounds on work extraction and erasure costs can be rederived from the presented framework.

We thank the Referee for their careful reading and for this detailed summary of our results.

Main Comments. 1. Can the previously introduced quantity c_ϵ (as defined in the supplementary notes) not also in principle be used for studying approximate instances of impossible tasks? The error ϵ for this is defined instead in terms of a distance measure.

This approach does not work in general, for the following reason:

The quantity c_ϵ provides the minimum cost of quantum channels in an ϵ -neighborhood of a specific quantum channel \mathcal{M} . But many information processing tasks are not canonically associated to a specific channel. Consider for example a single state transition $\rho \rightarrow \rho'$: in general, there are infinitely many channels achieving the transition with perfect accuracy, and in general these channels can have different costs. When a task is not uniquely associated to a quantum channel, one cannot simply pick one of the channels \mathcal{M} that achieve the task perfectly and search in an ϵ -neighborhood of \mathcal{M} , because the channel that has minimum cost for the desired value of the accuracy may be outside the ϵ -neighborhood of the particular channel \mathcal{M} one has picked.

As a concrete example, consider the task of transforming the maximally mixed qubit state $I/2$ into the Gibbs state $\Gamma = \frac{2}{3} |0\rangle\langle 0| + \frac{1}{3} |1\rangle\langle 1|$.

This task is achieved perfectly by the channel that maps every state into the Gibbs state Γ . This channel is a Gibbs-preserving operation and, as such, has zero cost.

But the state transition $I/d \rightarrow \Gamma$ is also achieved perfectly by the channel \mathcal{M} that maps

$|0\rangle\langle 0|$ into $\frac{2}{3} |1\rangle\langle 1| + \frac{1}{3} |0\rangle\langle 0|$

and

$|1\rangle\langle 1|$ into $|0\rangle\langle 0|$

This channel \mathcal{M} has a strictly positive cost. Hence, if we just pick channel \mathcal{M} and ask what is the cost of epsilon-approximating it, we will generally get a positive cost, whereas the cost of implementing the state transition $I/2 \rightarrow \Gamma$ (exactly or approximately) is zero.

In the revised version, this point is discussed in Supplementary Note 1.

So would the problem just be that the resulting quantity would be more difficult to work with and compute?

The main problem is the conceptual problem mentioned in the response at the previous point. Furthermore, using c_ϵ is not convenient when the desired task cannot be implemented perfectly, as in the case of quantum cloning. In this case, one could try to pick the optimal quantum channel \mathcal{M} that implements the desired task with maximum accuracy F_{\max} , and ask what would be the cost of implementing another channel \mathcal{M}' in an ϵ -neighborhood of \mathcal{M} . But even if the optimal channel \mathcal{M} is unique, this approach has the disadvantage that it uses two distinct approximation parameters (F_{\max} and ϵ) for the same notion (the accuracy in the execution of the desired task), and tightness of the bounds is generally lost in the combination of these two parameters.

Or does the quantity c in the manuscript have a conceptual advantage as well?

Yes, there is a fundamental reason why we consider the quantity c_0 instead of c_ϵ . The reason is that any implementation of an abstract information processing task always corresponds to a specific machine, described by a specific quantum channel \mathcal{M} . The minimum nonequilibrium cost that has to be paid in order to set up this specific machine is precisely the nonequilibrium cost of $c_0(\mathcal{M})$: the cost of the actual channel that has been implemented, not of some approximation thereof. Hence, the cost is given by $c(\mathcal{M})$, not $c_\epsilon(\mathcal{M})$. The problem, then, is to find a *direct* relation between the accuracy in the execution of a desired task and the nonequilibrium cost of the actual machine that has been used to implement that task. This corresponds to the settings adopted in our paper.

It is also worth mentioning that the approach of evaluating the accuracy operationally in terms of tests is in line with a general trend in other quantum applications, such as verification of quantum hardware, and the study of benchmarks for new quantum devices, which we mention in the revised version.

2. On a related point: under eq 10, it seems the comparison between bounds in the case of the transpose task is not fair – eq 10 is for a set of state transformations, not a channel.

Thank you for raising this point, we agree that a more careful discussion was needed here. In the revised version, we have rewritten this part, discussing the relation between our results and those in Ref. [33] more in detail and on a more even ground. The specific comparison in Eq. (10) of the earlier version has been removed, partly to avoid the issue mentioned in your report, and also because Eq. (10) did not appear explicitly in Ref. [33] and might confuse the readers.

Presumably one could apply eq 10 instead as follows: find a channel that implements the operation (say, transpose) to the desired accuracy, and then compute the α -Rényi relative entropies for the inputs and resulting outputs with that channel. Do the authors know how the resulting bound would compare with the one derived here?

In the revised version, this point is discussed extensively in Supplementary Note 10. There, we provide an explicit evaluation of the relative entropies (both the traditional quantum Rényi entropies and the “sandwiched” Rényi entropies introduced in more recent research) of the input and output states of a transposition task. Specifically, we consider the task of transposing qubit states that are either in the computational basis or on the equator of the Bloch sphere, and we pick the unique quantum channel that achieves the maximum transposition fidelity allowed by quantum mechanics. This channel has a strictly positive work cost, in agreement with a general bound now proved in the Supplemental Material. On the other hand, we show that the difference between the relative entropy of the outputs and the relative entropy of the inputs is always smaller than or equal to 0. These observations show that, in general, the difference in relative entropy between the input and output states associated to a given task does not detect the presence of a positive cost for the task under consideration.

3. I am uneasy with the terminology of “thermodynamic complexity”. Of course the word complexity has many different meanings across different fields, but it would be helpful if the quantity introduced here has at least some kind of connection with one of the established meanings. It is not clear to me why κ is much more than a measure of departure from equilibrium. As the authors say, in the classical case it is precisely that. The quantum case is more subtle, as shown by considering non-CP maps such as the transpose operation – but precisely what κ really measures beyond nonequilibrium is not explored very well. Could the authors give some more clarity on this and justify their terminology? Is there any relation with some recent notions of complexity relevant to thermodynamics, e.g. arXiv:2110.11371? (It appears not.)

We are grateful for these comments, which stimulated a valuable addition to the paper. We agree that (1) the term “thermodynamic complexity” was ambiguous, and (2) more insight into the definition was desirable. In the revised version, we addressed both issues. Regarding (2), we showed that our quantity κ measures the maximum accuracy allowed by quantum mechanics in the execution of a time-reversed task, defined in terms of a time-reversal of quantum operations introduced by Crooks and recently generalised in another work by one of us and coauthors. Accordingly, we replaced the name “thermodynamic complexity” with “reverse entropy,” thereby addressing (1).

Finally, we agree with you that there appears to be no relation between our entropic quantity and the notion of complexity considered in arXiv:2110.11371.

4. Similarly, apart from giving a useful computable bound to the cost c , what thermodynamical insight do we get from considering κ ?

Thank you for stimulating us to search for additional insights. The new interpretation of our thermodynamic quantity does indeed offer useful insights, establishing a connection between the cost of an information-processing task and the maximum accuracy of its time-reversal.

This insight solves the little puzzle you mentioned earlier in your report: what is the physical meaning of the nonzero κ of the transpose task? The explanation is that the time-reversal of transposition cannot be achieved perfectly in quantum mechanics: for example, in the fully degenerate case the time-reversal of transposition is transposition itself, and the maximum fidelity allowed by quantum mechanics is $2/(d+1)$, as shown by earlier works. The reverse entropy of transposition is then $\kappa = \log[(d+1)/2]$.

We can now understand deterministic classical computation and the transposition of quantum states in a unified way. For a classical deterministic computation, irreversibility arises when trying to compute a non-invertible function. In this case, irreversibility is directly related to a deviation from thermal equilibrium. For the transposition of quantum states, instead, the irreversibility has a physical origin: while transposition is mathematically an invertible function, it is forbidden by the laws of quantum mechanics. Due to this fact, irreversibility can take place even without deviation from thermal equilibrium.

Another insight arising from our new results is an explanation of why the nonequilibrium cost of classical and quantum cloning coincide. The time-reversed task of cloning is discarding, which can be achieved with the same accuracy both in the classical and quantum case. Hence, the reverse entropy assumes the same values for both classical and quantum cloning.

3. Is there any interesting relation with Ref. [30], which also characterises state transformations in terms of the min conditional entropy?

There does not seem to be a relation, at least not a direct one. The problems are different, as Ref. [30] provides necessary and sufficient conditions for the exact convertibility of a given input state into a given output state, while our work focuses on the approximate realisation of general information processing tasks.

In addition, the operators one computes the conditional min-entropy of are significantly different: in Ref. [30] (Eqs. (22) and (22)) one has to compute the min-entropy of a bipartite state involving the system of interest and a suitable reference frame. The input and target output appear as states of the system, and are tensored with suitable states of the reference. In our result, instead, one has to compute the conditional min-entropy of a bipartite operator where the inputs and target outputs are tensored with each other (not with the states of a reference). The presence of the conditional min-entropy in both works appears to be related to the fact that both papers use semidefinite programming for the optimisation of (certain subsets of) quantum channels, and this type of optimisation problems are generically connected to conditional min-entropies (see e.g. <https://iopscience.iop.org/article/10.1088/1367-2630/18/9/093053/>).

6. The writing is very good and should be accessible to a reasonably large readership. However the main text sometimes seems over-simplified to its detriment. For example, some quantities (including D_{\max} , H_{\min}) are never defined in the main text. For non-experts, definitions (perhaps left to the methods section) and some description of the meaning of the quantities, would be very helpful.

Thank you for the positive judgement on our writing, and for the recommendation to include the explicit definitions in the main text. In the revised version, all relevant definitions are included.

7. Statements about attainability of the bounds generally assume the ability to perform general Gibbs-preserving maps, rather than the better physically motivated subset of thermal operations. This issue is acknowledged by the authors; while it would be better to have a result better suited to thermal operations, this seems like a much harder problem.

Correct. Previous works generally used the term “work cost” in both settings, but, for the reasons you mention, we opted to introduce the term “nonequilibrium cost” for the cost of implementations using Gibbs-preserving maps, and we reserved the expression “work cost” to the cost of implementations using thermal operations.

The nonequilibrium cost provides a fundamental lower bound to the work cost. The achievability of this bound is generally nontrivial, but can be established in special cases. This is the case, for example, for all processes where the input and output systems are classical and fully degenerate. This category includes in particular the task of classical cloning, studied in our paper, for which we provide the exact value of the minimum work cost for every value of the accuracy. Another interesting example is the task of resetting quantum states to the ground state. In this example, our bound is achievable and we provide the exact expression for the work cost of approximate erasure for every possible Hamiltonian and for every possible value of the erasure fidelity.

Overall, I find that this manuscript addresses a well-motivated fundamental problem in the field of quantum thermodynamics, and is technically impressive, so certainly publishable.

Thank you for this positive assessment on the fundamental motivation and technical depth of the paper.

However, I do not see a strong conceptual advance over previous work. If I understand correctly, the definition of accuracy employed here (using observable expectation values rather than distance measures) results in a computationally more tractable version of the results from Refs. [31,32]. This appears to be the main contribution of the manuscript.

The results of Refs. [31,32] provided the cost of specific quantum channels, but in general did not provide the optimal tradeoff between accuracy and nonequilibrium. For example, many information processing tasks can be achieved by more than one quantum channel, and the results of Refs. [31,32] alone do not provide the minimum cost that has to be paid for a desired level of accuracy. Another conceptual advantage over Refs. [31,32] is that these works were limited to tasks that are perfectly achievable within quantum theory. Consequently, these previous results did not apply, for instance, to the task of quantum cloning, whereas this is possible (and done) here.

While computational tractability is definitely a bonus, the main contribution of our work is conceptual, and lies in the development of methods to assess the cost of accuracy directly in terms of information processing tasks, without passing through intermediate steps such as the inspection of specific quantum channels. This approach offers a tighter connection between information processing and thermodynamics, especially after the new insights included in the revised version, which relate the cost of accuracy to time-reversal of the information processing task under consideration. These concepts and methods represent a substantial advancement of the state of the art. In particular, and can be used to evaluate the thermodynamic resources needed in a number of quantum information tasks, as illustrated in the revised version.

Another substantial contribution of our work is to provide a rigorous framework for establishing the thermodynamic advantage of quantum machines over classical machines, based on measurement of the input and consequent reparation of the output. Our bound on entanglement binding machines provides a general way to establish such advantages. In addition, in the revised version we derived the minimum amount of nonequilibrium required by classical machines for storage and transmission of qubit states, thereby providing a fundamental bound on the classical work cost. Every machine that achieves the desired fidelity with an amount of work below the classical limit can then be certified to offer a quantum advantage. This result enables the demonstration of quantum-enhanced performance in noisy regimes where the existing quantum benchmarks cannot be applied.

While valuable, apart from this, it is not obvious what insight one can get from the introduced thermodynamic complexity.

This point is now addressed in the revised version, where the connection with reverse tasks provides new insights, and new applications (such as quantum storage, transmission, and erasure) that further showcase the usefulness of the techniques developed in the paper.

The examples studied are interesting but arguably not very useful since perfect cloning is anyhow an impossible task.

We respectfully disagree on this point. First, it is worth reminding that our results also apply to classical cloning, which can in principle be achieved perfectly. Our work establishes the minimum amount of nonequilibrium required by every classical replication process, in the realistic setting where the accuracy of the copies is not perfect. For fully degenerate systems, our result provides the exact value of the minimum amount of work required to copy classical information with a desired level of accuracy.

Moreover, while quantum cloning cannot be achieved perfectly, approximate cloning machines are important both for foundations (e.g. relations to the uncertainty principle, to the no-signaling principle, and to the precision limits of quantum metrology) and for applications (e.g. study of attacks in quantum key distribution). The relevance of quantum cloning to the community is attested by a vast literature, also including a full Review of Modern Physics dedicated to the topic. The thermodynamics of quantum cloning was largely unexplored so far, and our work establishes an important result in this direction.

Finally, it is worth mentioning that our results include not only cloning, but also all classical deterministic computations and their quantum extensions, which are ubiquitous in quantum computing. In this case, our result provides the exact amount of non-equilibrium resources needed in the realistic scenario where the implementation of the gates is imperfect.

In summary, I therefore do not see a convincing case for publication in this journal, unless the authors can provide some more remarkable insight or applications.

We appreciate your concern, and while we believe that original version of the paper already contained substantial advancements over the state of the art, we do agree that more insights and applications could greatly benefit the paper. In the revised version, we provide both insight (the relation between nonequilibrium cost and time reversal) and applications (storage, transmission, and erasure of quantum information). These applications are all relevant to the design of efficient quantum devices, for which they provide both the ultimate efficiency limits, and benchmarks that can be used to demonstrate quantum advantages in noisy scenarios.

Minor comments:

1. Perhaps consider altering the notation to make it clear that the quantities c and k depend on F ?

Thank you for this suggestion. It is correct that the quantity c depends on F , and the dependence has been made explicit in the revised version. The entropic quantity κ , however, does *not* depend on F , but only on the task itself. In the revised version, all the dependencies are made explicit.

2. Above eq 1: it may help for clarity to also state that invariance under time evolution means the state is diagonal in the energy eigenbasis.

The individual states do not have to be diagonal: only the projector on the input subspace needs to commute with the Hamiltonian. In the revised version, this precise formulation has been included.

2. In the first paragraph of methods: "fixed" channel presumably means the identity?

The fixed channel could be the identity, but it does not need to. Here "fixed" just means "independent of the process \mathcal{M} ". A clarification of this point has been included in the revised version.

4. Under eq 13 in methods: a bound is derived taking the observable as the projector onto the support of the output. When the output is mixed, $F = 1$ does not guarantee unit fidelity – does this show a downside of the approach used here?

The choice of the projector on the support is just an example of observable one could plug into our bound. Any other choice is equally valid, and every such choice would give a valid bound on the nonequilibrium cost.

As you correctly point out, here $F=1$ does not imply that the actual state is equal to the target state. $F=1$ is just a normalisation condition that guarantees that $\log F = 0$, and therefore the lower bound on the nonequilibrium cost is given exactly by the reverse entropy κ .

5. At many points in the supplementary notes, H is written instead of H_{\min} .

Thank you for pointing out this typo, which has been corrected in the revised version.

6. "Cloning" and "replication" seem to be used interchangeably – please clarify. The latter appears to be nonstandard terminology: does it simply refer to approximate cloning?

Both terms have been used interchangeably in the literature, see e.g. the original no-cloning paper by Wootters and Zurek, e.g. that uses “cloning” in the title and “replication” in the abstract, treating the two terms as synonymous. In the revised version, we nevertheless followed your suggestion, and preferred to use the word “cloning” throughout the paper.

REVIEWERS' COMMENTS

Reviewer #1 (Remarks to the Author):

I am satisfied with the changes that the authors have made in response to my previous report.

I recommend publication in Nature Communications. In my view the manuscript is in a good shape and can be published in its current form.

Although I recommend publication, I nevertheless have a somewhat tangential remark concerning one of the additions that the authors have made to the manuscript. I just mention this since it may potentially interest the authors, but it can safely be ignored. The remark concerns the mapping between states and observables that the authors introduce on p. 3, between equations (2) and (3). I would suggest that this map is closely related to Petz recovery channel [D. Petz, Sufficient Subalgebras and the Relative Entropy of States of a von Neumann Algebra, Commun. Math. Phys. 105, 123 (1986), D. Petz, Sufficiency of Channels over von Neumann Algebras, Q. J. Math. 39, 97 (1988), H. Barnum and E. Knill, Reversing Quantum Dynamics with Near-Optimal Quantum and Classical Fidelity, J. Math. Phys. (N.Y.) 43, 2097 (2002)].

Reviewer #2 (Remarks to the Author):

The authors have taken the comments from me and the other referee seriously and this is reflected in the revised manuscript. Apart from improving some terminology and notation, and answering some technical questions, the most significant change is a more thorough exploration of the physical meaning of one of the newly introduced quantities, the reverse entropy. They now show that this quantity can be expressed in terms of a suitably defined time reversal of the information processing task. This now links the thermodynamic cost of cloning, for example, to information erasure. They also further discuss work cost, and use their results to propose a benchmark on quantum communication channels in terms of their nonequilibrium cost.

I think these additions substantially improve the clarity and depth of the work. All questions from the other referee have also been satisfactorily answered. I therefore now feel able to recommend the paper for publication in this journal.

Response to Reviewers and List of Changes

We thank both Referees for their positive recommendation towards publication. A brief response to the Referee reports is included in the following.

In the current revision, we added references to Petz's recovery map, as suggested by Referee 1, and we addressed the list of editorial requests.

Response to Reviewer #1

I am satisfied with the changes that the authors have made in response to my previous report.

I recommend publication in Nature Communications. In my view the manuscript is in a good shape and can be published in its current form.

Although I recommend publication, I nevertheless have a somewhat tangential remark concerning one of the additions that the authors have made to the manuscript. I just mention this since it may potentially interest the authors, but it can safely be ignored. The remark concerns the mapping between states and observables that the authors introduce on p. 3, between equations (2) and (3). I would suggest that this map is closely related to Petz recovery channel [D. Petz, Sufficient Subalgebras and the Relative Entropy of States of a von Neumann Algebra, Commun. Math. Phys. 105, 123 (1986), D. Petz, Sufficiency of Channels over von Neumann Algebras, Q. J. Math. 39, 97 (1988), H. Barnum and E. Knill, Reversing Quantum Dynamics with Near-Optimal Quantum and Classical Fidelity, J. Math. Phys. (N.Y.) 43, 2097 (2002)].

We thank the Referee for their positive assessment of our revision and for the positive recommendation towards publication.

In the revised version, we have added a reference to Petz's recovery map, which is indeed related to the notion of time-reversal used in our paper (in fact, the relation was already mentioned in the references for the time-reversal, but we appreciate that it may be helpful to explicitly mention Petz's recovery map also in this paper).

Response to Reviewer #2

The authors have taken the comments from me and the other referee seriously and this is reflected in the revised manuscript. Apart from improving some terminology and notation, and answering some technical questions, the most significant change is a more thorough exploration of the physical meaning of one of the newly introduced quantities, the reverse entropy. They now show that this quantity can be expressed in terms of a suitably defined time reversal of the information processing task. This now links the thermodynamic cost of cloning, for example, to information erasure. They also further discuss work cost, and use their results to propose a benchmark on quantum communication channels in terms of their nonequilibrium cost.

I think these additions substantially improve the clarity and depth of the work. All questions from the other referee have also been satisfactorily answered. I therefore now feel able to recommend the paper for publication in this journal.

We thank the Referee for their appreciation of the effort put in the revision of our paper, and for the positive recommendation towards publication. We are especially glad that our discussion has been a true opportunity of scientific exchange, which greatly benefited the presentation and contents of the paper.